# Implementation Details for Controlling Contactless 3D Virtual Endoscopy

**Martin Zagar** [1,*]**, Ivica Klapan** [2,3,4]**, Alan Mutka** [1] **and Zlatko Majhen** [5]

1   RIT Croatia, WMC Department, 10000 Zagreb, Croatia; alan.mutka@croatia.rit.edu
2   Klapan Medical Group Polyclinic, 10000 Zagreb, Croatia; telmed@mef.hr
3   Faculty of Dental Medicine and Health, Josip Juraj Strossmayer University of Osijek, 31000 Osijek, Croatia
4   School of Medicine, University of Zagreb, 10000 Zagreb, Croatia
5   Bitmedix d.o.o., 10000 Zagreb, Croatia; info@avanza.hr
*   Correspondence: martin.zagar@rit.edu

**Featured Application: Our proposed solution is initially applied to three-dimensional virtual endoscopy, but it could be used by any kind of medical specialist that uses medical imaging.**

**Abstract:** In the medical world, with the innovative application of medical informatics, it is possible to enable many aspects of surgeries that were not able to be addressed before. One of these is contactless surgery planning and controlling the visualization of medical data. In our approach to contactless surgery, we adopted a new framework for hand and motion detection based on augmented reality. We developed a contactless interface for a surgeon to control the visualization options in our DICOM (Digital Imaging and Communications in Medicine) viewer platform that uses a stereo camera as a sensor device input that controls hand/finger motions, in contactless mode, and applied it to 3D virtual endoscopy. In this paper, we will present our proposal for defining motion parameters in contactless, incisionless surgeries. We enabled better surgeon's experience, more precise surgery, real-time feedback, depth motion tracking, and contactless control of visualization, which gives freedom to the surgeon during the surgery. We implemented motion tracking using stereo cameras with depth resolution and precise shutter sensors for depth streaming. Our solution provides contactless control with a range up to 2–3 m that enables the application in the operating room.

**Keywords:** contactless surgery; 3D visualizations; hand tracking

## 1. Introduction

Computer assisted surgery (CAS) and contactless surgery (CS) are not new concepts in modern telesurgery and telemedicine. We have had much experience with different approaches to resolving problems that occurred during the implementation of different frameworks and surgeon needs in the operating room (OR). By applying the concept of so-called personalized medicine, during the development of our original CAS and CS concept [1], expert members of our medical and IT teams created an enormous potential for transforming diagnostic and therapeutic procedures into the best possible solutions. However, does our CS in rhinology really represent the future of smart surgery? Do we really have real technological advances in rhino surgery in our hands, as the great Professor Heinz Stammberger once said?

The visualization and exploration of 3D medical data (CT, MIR, or ultrasound) have great importance in real medical applications. Medical visualization system users are primarily medical doctors from a specific discipline, such as radiology, surgery, nuclear medicine, or radiation treatments, who use such tools for diagnostic support, treatment planning, intraoperative navigation, etc. [2–4].

This paper focuses on improving the "In the Air" doctor–computer interaction, or HCI (human–computer interaction), during surgery in the clinical environment. The rest of

the paper is organized as follows. In Section 2, we set the problem of navigation through the human body. In Section 3, we describe our input modalities for surgeon–computer interaction and motion recognition methods used for controlling the contactless 3D virtual endoscopy (VE). In Section 4, we present our results. In Section 5, we discuss the potential of our solution, with conclusions in Section 6.

In modern OR, the surgeon, as a 21st-century man, thinks differently, with a new visualization aspect and understanding of the ecosystem, visualization space, and self-and anatomy-awareness of his patients. Our approach could be an important step towards the strategy of enhancing surgeons' capacities and increasing their overall satisfaction and precision since we enable the integration of real and virtual objects in the surgical field. We started our initial research by applying virtual reality concepts in our first Tele3D CAS in rhinology where we implemented a new framework for the transfer of computer data (images, 3D models) in real-time during the surgery and, in parallel, of the encoded live video signals that were already presented in [5]. We demonstrated this approach with an example of 3D-computer assisted navigation surgery of the nose and paranasal sinuses with simulation and planning of the course of a subsequent endoscopic operation per viam virtual endoscopy, which overcomes some difficulties of conventional endoscopies, such as "standard" functional endoscopic sinus surgery (FESS) or Tele-FES.

## 2. Materials and Methods

Lately, complex combinations of different computer-assisted features are greatly helping surgeons' "space problems" in the OR, but surgeon–computer interaction is also getting more complicated for the surgeon, based on our experience of implementing different approaches for contactless control and the hand-gesture touchless surgeon–computer interface in different operating rooms at Klapan Medical Group Polyclinic and Agram Special Hospital that were also discussed in [6]. From this point of view, augmented spatial anatomic elements of the human anatomy, simultaneously combined with the use of 3D medical images, already presented in [7] and 4D medical videos, could be combined with touchless navigation in space and such complex data should enable higher intraoperative safety and reduce operating times, as proven in [8]. In those references, we proved the possibility of touch-free operations on medical datasets, and we extended this using motion tracking for the precise control of virtual movements. By this approach, we achieved medical dataset management with possible actions of rotation of the whole dataset or just part of it, cutting and spatial locking within the dataset, measuring regions of interest, as well as slicing through datasets.

The use of medical models so far for the purpose of obtaining a better "understanding" of the operating field [9], as well as the application of unused IT technologies in the OR, has undoubtedly enabled surgeons to better "understand" human anatomy and the "human space" problem [10]. On top of that, 2D-MSCT (two-dimensional multislice computed tomography) and MRI (magnetic resonance imaging) datasets are displayed in 3D models, allowing the operating surgeon to "enter" and move around them per viam our CS [7].

*How Can We Set the "Space Problem" in Modern Contactless Surgery?*

So, what is currently "missing" in the standard OR? Why is contactless image management so important? By applying CS in [5], we previously enabled exactly what was missing in advanced virtual reality and 3D computer-assisted technologies in Tele-3D-CAS, and with this research, we updated and added missing functionalities for routine operations and teleoperations. From that point of view, anatomy could be examined from all points of the surgeon's view, as follows:

- Surgeons can analyze, interact with, and detect the region of interest (ROI) in pre-operative planning;
- Interaction is enabled by using a stereo-depth camera for tracking (we used Intel RealSense D435 Camera), so we can simulate a patient's surgery in the same environment before the real surgery;
- During the surgery, CS allows the surgeon to visualize and analyze the current performance of the surgery;
- It is possible to view the simulation of the complete surgery and the outcomes from the surgeon's point of view [11];
- It is possible to see the outcomes by simulating what might happen during some procedures of removing or cutting some ROIs from a different point of view of surgeons, providing the surgeon new opportunities for diagnostics and surgical planning, and preparing the surgeon for outcomes in key situations.

### 3. Input Modalities and Methods for Interaction with Computing Devices

Computers are equipped with various input devices/sensors to allow them to receive information from humans. Standard input devices have a standardized communication method with the computer (keyboard, mouse, touchscreen) and afford practical adjustments to the user. Specific modalities can provide a richer interaction depending on the context (motion detection, speech recognition, etc.).

The latest 3D imaging cameras (we used Intel Realsense 435 series for this research) offer fantastic opportunities to redefine how to interact with computing devices ultimately. The camera provides fast 3D scene acquisition, which allows for tracking body position and gestures in real-time. To design a successful software application, the strengths of each used modality must be understood. The final goal is to make it a natural but exciting experience for the user and minimize fatigue of the hands, fingers, or voice. Sometimes, multiple modalities need to be included because some of the users may prefer specific modalities over others. Here, we analyzed traditional input methods so we could evaluate which kind is most appropriate for surgeons in the OR:

- Hands: Mid-air hand gestures allow for dynamic and engaging interaction with 2D and 3D objects. They also allow more comfortable, more literal direct manipulation. One of the negative aspects is that they can be tiring over long periods, and the precision is limited.
- Face: 2D (image) and 3D (point cloud) data analysis of the human face provide the best information about emotions, natural expressions, and engagement. The negative side is a large variability of expressions across ages, cultures, and personalities.
- Speech: Human language is the most potent means of expression. Humans generate different sounds in different situations, and it is beneficial to integrate speech control with some other modality. Negative aspects are environmental noise and social appropriateness.
- Touch: Very used and highly optimized on modern mobile devices and touch screens with reasonably good precision. The downsides are limited interaction in 2D space only.
- Mouse: This allows an accurate indication of a 2D point. Manipulation is more challenging in the 3D world.
- Keyboard: This allows quick and accurate text entry. Keyboard shortcuts can be quick escapes but are not intuitive.

### 4. Results

During the surgery, based on our experience, we found that the best modality for surgeon-computer interaction is contactless interaction by hand. The next issue we needed to resolve was the ability for real-time interaction on a medium-range computer system (due to the cost issues). We used a disposal computer system with 4-core Intel Ice Lake processor i5 series, 1.5 GHz Core speed, 8MB L3 Cache memory, $4 \times 1280$ KB L2 Cache memory, and $4 \times 48$ L1 Data Cache memory; 8 GB DDR4 RAM and Intel Xe built-in graphics. For

capturing the hand motions, we used an Intel RealSense D435 stereo vision camera that has the depth perception capability needed for capturing 3D hand and finger motions. Since our computer system specification did not allow adding any external graphics solution and computer power, we focused on lightweight hand-tracking software solutions. Since MediaPipe Hands [12] enables precise hand and finger tracking solution that does not require any resource-powerful computer system for hand tracking solutions and is based on machine learning (ML) software, we implemented and optimized that solution. MediaPipe Hands 213] used in the API utilize an ML pipeline consisting of:

- a palm detection model for defining the ROI in the picture captured and returning a hand bounding box, as shown later in Section 4.2;
- a hand landmark that operates on the cropped region of interest defined by the palm detector and returning precise 3D hand keypoints.

To detect initial hand locations, a single-shot detector is used. This approach uses a single deep neural network that can detect left and right various hand sizes and even can detect hands overlapping. In hand detection, the lack of high contrast patterns (like for example eyes on the face) makes it comparatively difficult to detect them reliably from their visual features alone. Instead, providing additional contexts, such as arm, body, or person features, aids accurate hand localization. The presented model achieves an average precision of 96% in palm detection. We measured this rate by re-initiating the model with different hands and initial positions for tracking in a row of 100 attempts with four people (eight different hands) serving as a testbed. All four failed attempts occurred when the palm was too near to the person's head. We indicated a successful attempt with the successful palm detection within three seconds from the initial positioning of the hand in front of the camera, and detecting the palm in the region of interest frame (more details in Section 4.2).

We implemented MediaPipe 21 Hand Landmarks for precise hand detection by identifying 21 3D hand-knuckle coordinates [13]. The implemented MediaPipe algorithm provides up to 20 FPS hand/gesture detection on a provided system. The outputs from the algorithm are:

- Center of hand in 2D Image Pixel coordinates;
- Center of hand in 3D world/camera coordinates;
- Each finger state (open or closed): thumb, first, second, third, and fourth finger.

We implemented the following gestures: ONE, TWO, THREE, FOUR, FIVE, FIST, YEAH, ROCK, SPIDERMAN, and OK, as shown in Figure 1. Each of the gestures is defined by a finger state. The full list of gestures is presented in the following section.

### 4.1. Full List of Motions Recognized by the Systems

The following gestures were implemented/tested within the developed system:

- ONE: the first finger is open
- TWO: the first finger and the thumb are open
- THREE: the first, second finger, and the thumb are opened
- FOUR: the first, second, third, and fourth fingers are opened
- FIVE: all five fingers are opened
- FIST: all fingers are closed
- OK: the first and the thumb are connected, other fingers are up
- YEAH: the first and the second fingers are opened
- ROCK: the first and the fourth finger are opened
- SPIDERMAN: thumb, the first and the fourth fingers are opened.

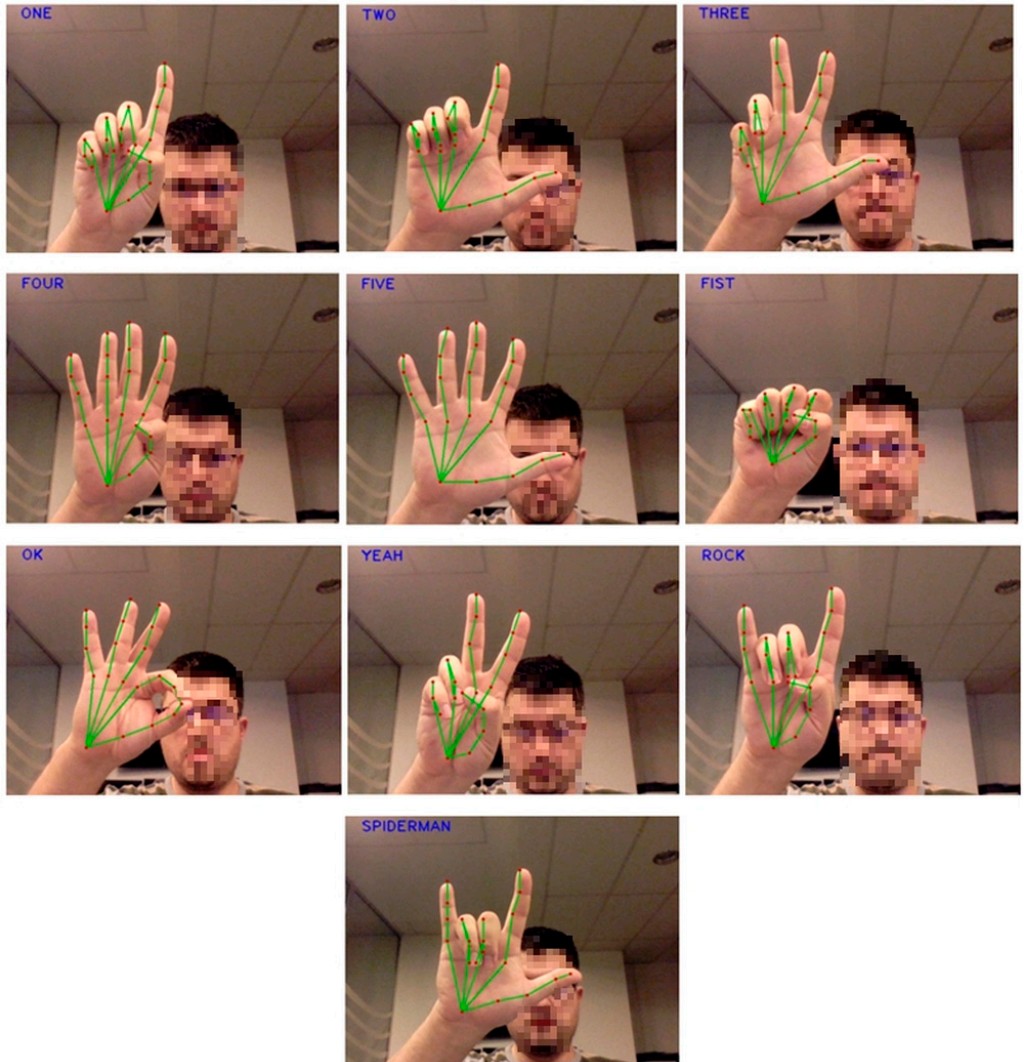

**Figure 1.** Motions recognized by our solution.

### 4.2. Visual Feedback

Feedback on visual interaction provides important information for the contactless hand gesture control system. In order to enable all the features, it is necessary that the surgeon understands how to control our application. Several requirements/program features were implemented in our user interface, as shown in Figure 2, to enable fast feedback and accurate and precise interaction with the system:

- UI that considers human ergonomics: We created ribbon-based UI that can be controlled contactless during the surgery and in the surgery planning.
- Fast and informative visual feedback: Within 100 ms we provide the info on what happened and the surgeon can see what are the next options.
- Intuitive and clear visual designs and text feedback.
- Visual feedback for the optimal user distance from the camera ("Optimal distance", "Move Closer", "Move Back").
- User viewport: A small screen where the surgeon can see what the camera detects.

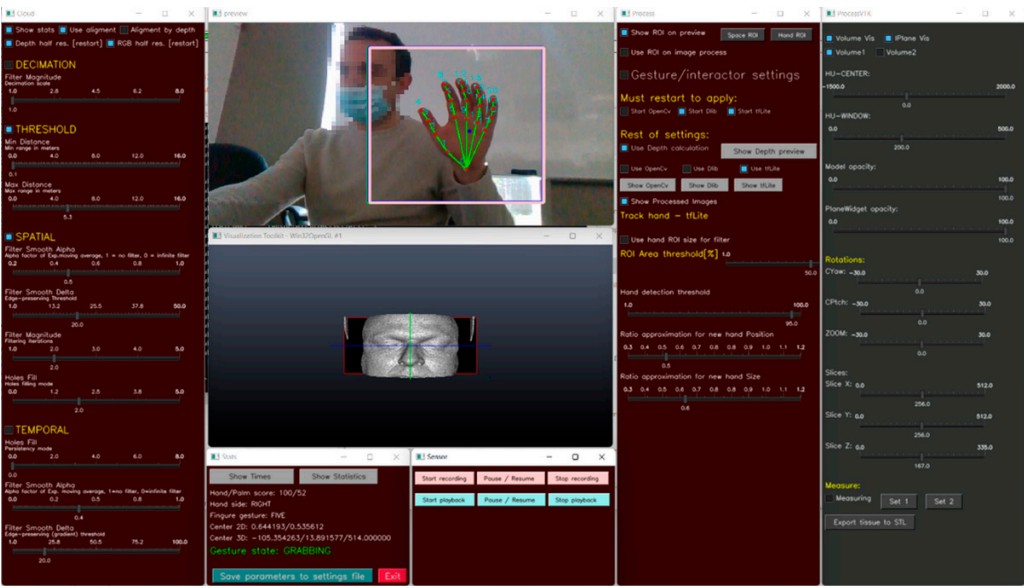

**Figure 2.** User Interface.

Control Ribbons 1 and 2 shown in Figure 3 can control different parameters. Control Ribbon 1 sets spatial and temporal filtering and threshold for depth dimension. For temporal and spatial filtering, we used medium smoothing of Alpha and Delta channels to preserve edge and corner detection of the hand. Values of these filters depend also on the background light. For darker backgrounds, these filters should be adjusted to higher values. The Filter Smooth Alpha channel for indicating the movements should be adjusted on values above 0.7 when there is less than 300 lux in the room. Similarly, the Filter Smooth Delta channel for edge-preserving in the same conditions should be above 30. The adjustments are usually made only for the initialization of the software in the runtime environment since the system once installed stays in the OR.

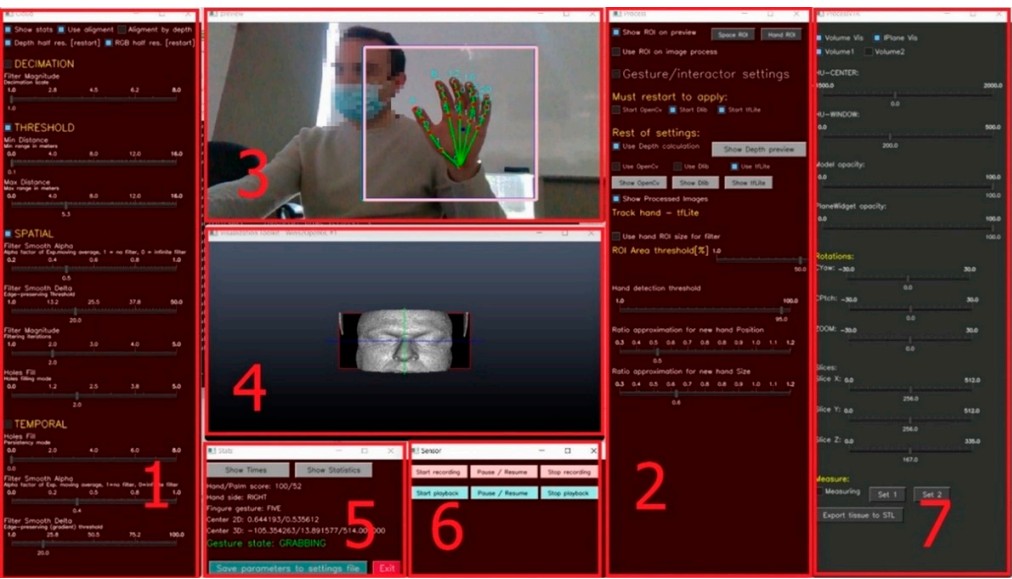

**Figure 3.** Control Ribbon.

Control Ribbon 2 serves for the Gesture settings and defines the region of interest where the hand will be tracked (this is the purple rectangle shown in Ribbon 3). Adjusting the size of the region of interest is important to focus on the hand of the surgeon in control, removing all unnecessary background noise and other background movements. Ribbon 3

shows the view of the user and how the camera interacts with the user. Ribbon 4 represents the visualization of the medical data. This ribbon could also be put in full-screen mode, once all the initial adjustments are performed.

While the hand moves on motion Five, the system is engaged, and moving the hand will move the data visualized in Ribbon 4. Different options for visualization could be engaged with initializing the motion Two to choose between the options for visualization, as shown in Figure 4. Options for the visualization of medical data include changing the center of visualization and changing the size of the window, volume opacity, plane opacity, camera pitch, and setting the 3D slices coordinates. The main mode of control is visualization in virtual endoscopy where the current visualization dataset could be fully moved and controlled.

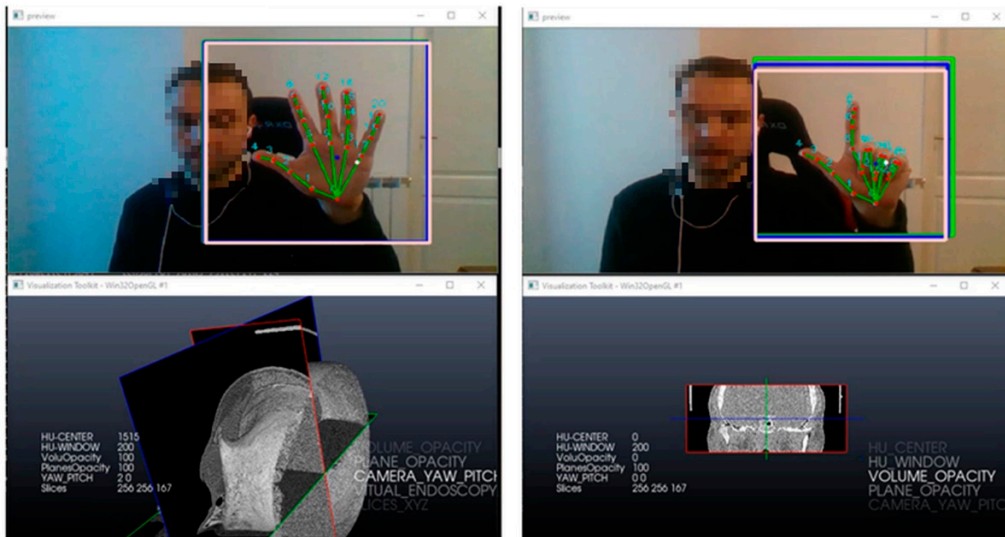

**Figure 4.** Controlling the visualization options.

Ribbon 5 shows what the system can read and the current gesture state and parameters for setting the 2D centers for the planes and overall 3D center. Ribbon 6 gives an option to record the visualization sequence for later surgical planning, while Ribbon 7 represents feedback on the current position, zoom options, center of the planes, and opacity modes.

## 5. Discussion

This section is dedicated to our cover destination: what we need to know about CS, what we need to do, and where to stop? Maybe, when we achieve a "collective mind" by enabling an infinite number of data sources (various CS-data bases) [13], the newest theory which led to the development of swarm intelligence applications. By connecting to a "collective mind" or "collective knowledge" per viam CS, we will be able to share all knowledge and skills from a centralized source [14].

Contact navigation surgery in OR (CAS, NESS, etc.), inaugurated as an elite surgery worldwide in the early 1990s, today represents the standard part of the equipment of each slightly better operating room. This form of navigation must enable the surgeon to recognize the spatial position of surgical instruments in the operating field, as well as a complete "understanding" of comprehensive 3D human anatomy during the surgical operation itself.

However, when using this form of surgical navigation, each surgeon is forced to "translate" 2D MSCT/MRI data into understandable 3D shapes countless times during the operation. His consciousness must repeat countless times, but also correctly understand this mapping of two different worlds, which was our primary goal when designing the system. This much-needed process, which takes place throughout contact navigation surgery, sometimes makes it difficult to safely perform surgery, primarily due to possible

"fatigue" of the surgeon, and thus his reflexes, and above all safe determination of depth and dimensionality of 3D space. During endo-operations that last longer, there may be uncertainty in choosing the correct implementation of the operation, which is addressed with our high-fidelity UI/UX.

We implemented our system in two operating rooms and asked several medical specialists to test the system and give feedback, so overall it was applied during eight different endoscopies. We compared the time needed to obtain the data regarding the maxillary sinus in standard visualization with the help of a medical assistant for controlling the visualization, and with our system for virtual endoscopy. We tried the system in two different environments, with high and low background lights, and adjusted appropriate Alpha and Delta channels to preserve edge and corner detection of the hand. Results are presented in Table 1.

**Table 1.** Comparison of the time needed for visualization of the maxillary sinus from the initial outer view of a skull with our proposed system for virtual endoscopy and standard procedure with the help of a medical assistant.

| Attempts | The Proposed System (in Seconds) | Standard Procedure (in Seconds) |
|---|---|---|
| Attempt 1 (Medical Specialist 1; 250 lux background light; Alpha ch. = 0.8; Delta ch. = 35) | 4.5 | 8.9 |
| Attempt 2 (Medical Specialist 1; 500 lux background light; Alpha ch. = 0.5; Delta ch. = 20) | 4.4 | 9.1 |
| Attempt 3 (Medical Specialist 2; 250 lux background light; Alpha ch. = 0.8; Delta ch. = 35) | 5 | 9.5 |
| Attempt 4 (Medical Specialist 2; 500 lux background light; Alpha ch. = 0.5; Delta ch. = 20) | 5.4 | 10 |
| Attempt 5 (Medical Specialist 3; 250 lux background light; Alpha ch. = 0.8; Delta ch. = 35) | 3.7 | 8.7 |
| Attempt 6 (Medical Specialist 3; 500 lux background light; Alpha ch. = 0.5; Delta ch. = 20) | 4.4 | 8.9 |
| Attempt 7 (Medical Specialist 4; 500 lux background light; Alpha ch. = 0.5; Delta ch. = 20) | 4.4 | 9.1 |
| Attempt 8 (Medical Specialist 4; 250 lux background light; Alpha ch. = 0.8; Delta ch. = 35) | 5 | 9 |

In addition, the CS concept allows for the visualization of any anatomical object by looking at the same subject from different points of view, giving it 3D dimensionality, which we already proved in [15]. The difference in perspectives is used to generate a depth map of hands and motions by calculating the distance from the camera sensor to every pixel in the scene and ROI [10]. In order to use the depth and motion tracking camera properly, we needed to define some standard moves of the surgeon's hand to speed up the process of motion and depth recognition and to have the more precise real-time outcomes we stated in [16].

With this approach, we can achieve our contactless surgery by using the controls of depth and motion tracking cameras and with the fully operable list of virtual movements reflecting standard moves through patient datasets, such as rotation, cutting, spatial locking, measuring, and easy slicing with virtual movement and medical modeling accuracy. So, CS's beauty attracts surgical beauty [8].

## 6. Conclusions

In the present medical world, without touching the screen during the contactless visualization of medical datasets [2], we enable virtual diagnostics and surgery of 3D spatial data [3], and we can shape the visualization in a way that does not exist in reality.

We also proved that it is possible to use technology to make something that we cannot see in the real world, visible to doctors/surgeons, i.e., to experience and understand what we cannot realistically see, or what does not really exist as shown in touch-free navigation through radiological images presented in [4].

In this way, we "understand" the 3D spatial relationships of the operating field as they are in human beings. So, with our plug-ins, we give CS the chance to be seen and have its story told with interaction with medical data through our interface. This is some kind of surgical "fly" with CS! Surgeons reported that the interface was intuitive and easy to use. Removing the man-in-the-middle in interaction with the visualization system by controlling the contactless system speeds up the whole process by on average 50% and removes any interaction with a medical assistant who was in charge of controlling the visualization system during the surgery. Moreover, surgeons reported less stress during the navigation because they felt that everything is under their control.

In conclusion, the use of augmented spatial anatomic elements that simultaneously overlap the control field to visualize 3D medical datasets in a contactless way enabled higher intraoperative safety, reduced operating times, and provided higher fidelity through our user-friendly interface.

**Author Contributions:** Conceptualization, M.Z. and I.K.; Formal analysis, M.Z.; Funding acquisition, M.Z.; Investigation, I.K. and A.M.; Project administration, M.Z.; Resources, M.Z. and I.K.; Software, A.M.; Supervision, Z.M.; Validation, M.Z., I.K. and Z.M.; Visualization, A.M.; Writing—original draft, M.Z.; Writing—review & editing, I.K. All authors have read and agreed to the published version of the manuscript.

**Funding:** This research was funded by EIT Health RIS Innovation 2020 Grant.

**Informed Consent Statement:** Informed consent was obtained from all subjects involved in the study.

**Conflicts of Interest:** The authors declare no conflict of interest.

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
