# Peer review of "Implementation Details for Controlling Contactless 3D Virtual Endoscopy"

_applsci, doi:10.3390/app12115757_

Round 1

Reviewer 1 Report

In this manuscript, the authors show a way to connect gestures with commands of an endoscope, aiming at facilitate contactless surgeries. As the core part of this manuscript is modality recognition, the authors should highlight the improvements they make, instead of simply citing mature algorithms. The manuscript looks more like an application note, thanks to the technical details, but not scientific research.

Assuming the hardware and software the authors use are the most appropriate, important information is still missing. In Line 162, the authors mention that additional contexts are needed for accurate palm detection. How important these contexts can be and how long will it take to adjust the contexts if palm localization fails? The authors only mention a successful rate of 95.7% without stating the way of testing. Similar content is also missing in Line 214 and 222. The authors mention the influence of background light without solid data.

For the format of this manuscript, if the first author and the third author come from the same organization, I assume they ought to share the same superscript number. Full spelling is needed for the DICOM in Line 17. Quotation marks are in the wrong place in Line 51 and 99. Typos can be seen in Line 174 and 231.

Author Response

In this manuscript, the authors show a way to connect gestures with commands of an endoscope, aiming at facilitate contactless surgeries. As the core part of this manuscript is modality recognition, the authors should highlight the improvements they make, instead of simply citing mature algorithms. The manuscript looks more like an application note, thanks to the technical details, but not scientific research. -> We added additional info about the performance and improvements in the conclusion part.

Assuming the hardware and software the authors use are the most appropriate, important information is still missing. In Line 162, the authors mention that additional contexts are needed for accurate palm detection. How important these contexts can be and how long will it take to adjust the contexts if palm localization fails? The authors only mention a successful rate of 95.7% without stating the way of testing. ->We elaborated on how we tested and provided more info on the positioning context.

Similar content is also missing in Line 214 and 222. The authors mention the influence of background light without solid data. -> We provided more info on parameters related to background light (Filter Smooth Alpha and Delta channels).

For the format of this manuscript, if the first author and the third author come from the same organization, I assume they ought to share the same superscript number. -> This is changed.

Full spelling is needed for the DICOM in Line 17. -> Explanation is added.

Quotation marks are in the wrong place in Line 51 and 99. -> Quotation marks are changed.

Typos can be seen in Line 174 and 231. -> This is also changed, thanks.

Reviewer 2 Report

Please find the comments in attachment file. 

Author Response

  1. No, the camera does not have any special feature itself, it just a class of stereo cameras with depth motion tracking. It should be mountable and with depth streaming. You can find multiple solutions on the market.
  2. References are added.
  3. Summarized in a new paragraph.
  4. This is changed. Now, this is a subsection in the section on Materials and Methods.
  5. Bullet points are capitalized.
  6. Added a new heading for Results.
  7. Added a new heading for Conclusions.
  8. Changed errors in two references [13] and [15].

Reviewer 3 Report

The article is extremely interesting as it presents the creation of  an additional feature to the already fast developing new technologies used in surgery. The authors bring novelty to the subject and this article is worth to be published, after some  concerns are addressed.

  1. in the abstract the term “surgery in the air” appears, this is rather confusing as some people may understand that it refers to flight surgeons (which is not the case).
  2. Conclusions are not clearly evident in the text
  3. Self- citing  can raise some concerns. While being impressed by the amount of published articles on the topic the authors have, the readers should clearly be able to read and understand what was already published and what is new information. Also a more neutral manner.
  4. Page 8, rows 278-283 - the idea of the paragraph is not clear, please rephrase it
  5. The article ends with the following statement: “definitely enabled higher intraoperative safety, reduce operating time and provide higher fidelity through a user-friendly interface”. It is not clear how the testing was done and what were the results of the testing so this statement is clearly justified (eg. How many interventions, average time of intervention, what were the differences compared to the control group; how was surgeon’s satisfaction measured, etc.)

Author Response

  1. This is changed to contactless surgery
  2. and 5. The conclusion section is updated with additional data collected from surgeons' feedback and the time needed to perform some standardized action
  3. We tried to show that we have experience in this research field and that we continuously improve the system and the approach. However, we tried to make it more neutral in several references, for example: "We started our initial research by applying virtual reality concepts in our first Tele3D CAS in rhinology where we implemented a new framework for the transfer of computer data (images, 3D-models) in real-time during the surgery and, in parallel, of the encoded live video signals that was already presented in [2]."
  4. The paragraph is rephrased to make it clear.

Round 2

Reviewer 1 Report

The authors have answered my questions properly. I have no further comments. 

Author Response

Thank you

Reviewer 2 Report

No

Author Response

Thank you

Reviewer 3 Report

I have seen a considerable improvement in the content of the article, as most of my concerns have been addressed, however there still is one problem which was only partially solved: the implementation. It is still not very clear which were the results when tested in real life setting (OR). I have seen the data introduced in the conclusion part, but I believe it needs to be mentioned in the results section. Also the testing  results need to be presented in more details. For eg the reduction in the time needed for the procedure to be performed vs standard procedure time (in numbers) would be of interest, this data can also be presented in a separate table.

I am looking forward to seeing the final form and reading it in the journal.

Author Response

Thanks. We added in the results section comparison of the time needed for visualization of the maxillary sinus from the initial outer view of a skull with our proposed system for virtual endoscopy and standard procedure with the help of a medical assistant, as a table with implementation details.